# Self-Reported Flow in Online Learning Environments for Teacher Education: A Quasi-Experimental Study Using a Counterbalanced Design

Lionel Alvarez [1,2,*] , Romaine Carrupt [3], Catherine Audrin [4] and Philippe Gay [5]

1   CERF, University of Fribourg, 1700 Fribourg, Switzerland
2   CRE/ATE, Haute École Pédagogique, 1700 Fribourg, Switzerland
3   Haute École Pédagogique Valais, 1890 St-Maurice, Switzerland; romaine.carrupt@hepvs.ch
4   UR MI, Haute École Pédagogique Vaud, 1014 Lausanne, Switzerland; catherine.audrin@hepl.ch
5   UR EN, Haute École Pédagogique Vaud, 1014 Lausanne, Switzerland; philippe.gay@hepl.ch
*   Correspondence: lionel.alvarez@edufr.ch

**Abstract:** Digitization in teacher education is currently being promoted, but the choice between face-to-face instruction and online learning environments remains challenging. Previous studies have documented ambivalent results regarding personal preference and academic achievement, and experimental investigations into attention comparing learning in these two settings are largely lacking. In this context, the present study adopts a counterbalanced design to compare different dimensions of student experience of flow in face-to-face settings and online learning environments. Two groups of students in teacher-training programs (*n* = 37) completed an *EduFlow* questionnaire at the end of the same interactive courses in the two different settings. The results indicate globally lower attention and engagement in the online environment, suggesting that in-person instruction induces better cognitive absorption, greater time transformation, and a stronger autotelic experience. While the findings represent a contribution to the discussion on how to best design online education, more research is needed to identify the specific mechanisms regarding attention and motivation that can impact flow in these two environments.

**Keywords:** distance learning; online classroom; self-reported optimal learning experience; *EduFlow*; counterbalanced design





## 1. Introduction

Teachers are responsible for managing the information they impart to students, but also for the ways they transfer this information. The specific environment—digital or face-to-face—undoubtedly has an impact on both the quality of learning and the quality of student engagement. At present, instructional design encourages blended learning formats for the flexibility they offer in terms of location, actions, mediations, and other pedagogical considerations [1]. For this reason, it can be observed that blended learning, defined as "a formal education program in which students learn part online, part away from home, and along a learning path" [2], is developing rapidly at universities for teacher education. It is frequently argued that this type of instruction promotes student participation and that it is congruent with a constructivist vision of learning rooted in student-centered teaching. In addition, it is often recommended that teacher-training institutions remain up to date on the latest pedagogical methods, as it is important to model good practices for future teachers. As a rule, teacher education programs employ blended learning, i.e., part face-to-face instruction, part distance learning; however, during student-teaching internships, the experience is limited entirely to face-to-face interactions in the classroom.

Impacts of blended learning in teacher education programs, as well as student perceptions thereof, constitute a complex issue that several studies have already documented with the following findings:

- There is no difference between face-to-face learning and blended learning in terms of academic achievement [3];
- Knowledge transfer is stronger in blended learning groups [3];
- Taking individual differences into perspective, having an extroverted personality appears to correlate with higher satisfaction in blended learning settings [4];
- In a retrospective study, student participation in in-class settings was higher than in online activities [5];
- Students appreciate the flexibility offered by online activities [5];
- Students perceive a higher affordance of online material [6];
- The set-up of the learning management system has a positive influence on student participation [6]; and
- Students generally prefer face-to-face interactions and traditional modes of learning [7].

These initial findings indicate that blended learning in teacher education is not a panacea, but rather a teaching option that has potential benefits if used wisely. In addition, it is now unlikely that universities for teacher education will want to avoid online learning due to developments regarding aspects such as scalability, accessibility, and focus on digital skills development [8,9]. These results invite instructional designers to introduce blended learning into their teaching and to think critically about how to combine learning environments in the specific field of teacher education.

In addition to these empirical findings on learning and student satisfaction, however, the research suggests that digital devices also have potential harmful impacts on well-being [10], physiological variables [11], and cognitive processes [12]. Although these factors are generally important in learning contexts, they are of particular relevance in teacher education, which must be efficient in terms of skills development while also imparting effective practices that can later be transferred into the classroom when students become teachers. Indeed, one method of teaching is to model practices, and instructors at universities for teacher education are consequently encouraged to create well-designed learning sessions, both for teaching efficacy and for the models of teaching they embody. Therefore, it is particularly important to focus on evidence-based interventions when making decisions about instructional design. A major question concerns whether future teachers perceive the online learning experience in a similar way as they do face-to-face instruction. The answer has the potential to support designing blended learning settings.

### 1.1. Synchronous Online Learning Environment

When synchronous remote communications become necessary between students and between instructors and students, online settings are an option. Several digital platforms have been available to teachers and universities for some time, and these digital tools experienced a boom due to the health and safety measures in place during the COVID-19 pandemic. While studies have long proven that students experience these types of learning environments as conducive to expressing their opinions in groups [13], online learning models are also perceived as an interesting opportunity to support interactivity, to reach students who cannot participate in person due to geographic distance, and to build communities [14]. A variety of formats [15] could help to promote student engagement online; examples include webinars, use of video, online discussion boards, wikis and blogs, gaming, and group projects. Interestingly, teachers assess strategies to promote engagement as more important than students do [16], with students placing less value on icebreaker discussions, virtual lounges, collaborative activities, and other interaction-related strategies.

When choosing blended or distance learning over traditional, synchronous in-class instruction, it is critical that pedagogical goals and advantages are clearly identified, as there are significant challenges surrounding instructional design for online teaching [17]. In a systematic literature review [18], various technical, organizational, and pedagogical barriers were identified that must be overcome to enable effective implementation of online classrooms. For instance, digital literacy, misunderstandings, or students' level of self-motivation need to be specifically considered. Moreover, the development of online

materials should be supported by evidence-based practices or guidelines that have been validated through empirical findings [19–21], making it possible to select media resources for a specific intention; ideally, the blended learning experience should be backed up by a detailed design process to ensure an optimal learning experience (e.g., the Shewhart cycle for blended learning [22]). In addition, issues surrounding student cognitive load may pose problems; to best foster learning, online instructional designers are called on to include specific strategies to mitigate the negative impacts of digitized education [23].

Overall, distance learning expands the possibilities in instructional design by offering an environment that can be built from existing resources, adapted, or newly created. It represents a viable alternative to in-person teaching that, although widely discussed in the relevant literature, has rarely been examined with a detailed focus on optimal learning experience from the student perspective.

### 1.2. Flow as an Optimal Learning Experience

With relation to the popular concept of flow [24,25] as studied in the context of general functioning [26], an optimal learning experience is defined as a "psychological mental state of a person who is immersed in an activity with energized concentration, optimal enjoyment, full involvement, and intrinsic interests, and who is usually focused, motivated, positive, energized, and aligned with the task at hand" [27]. A four-dimensional model to describe the experience of flow in educational contexts has been proposed [28]:

- Cognitive control or increased concentration and immersion in the task when learners are so intensely involved in a task that they feel completely absorbed.
- Time transformation or alteration in the perception of time, sometimes leading to a lengthened duration of immersion in the task, when the learner experiences a loss of time reference or is unable to correctly assess the length of time they stayed engaged.
- Loss of self-consciousness or lack of self-concern related to an increase in importance of the psycho-social dimension of learning when learners forget other needs because of total engagement in a task.
- Autotelic experience or well-being during task performance resulting from purpose in the task itself that enhances persistence and the desire to engage in the activity again; when learners feel good, balanced, and in tune with the present moment, so that they wish to repeat the experience.

Previous findings indicate that the field of study has an influence on the student experience of learning in online activities [29,30]. For instance, several studies have already investigated blended or distance learning environments in teacher education. In a 2003 study, Khine and Lourdusamy [31] described learner satisfaction reported in a blended learning experience in Singapore and proposed that online participation could be incentivized by grading online activities. Donnelly [32] examines blended problem-based learning in a teacher education program in which the online environment was mainly designed for collaborative work following face-to-face activities, and asynchronous tasks targeted aspects such as organization, synthesis, and critiques with regard to the synchronous tasks. In this case, no specific encouragement was needed to promote participation in the online environment, possibly because it was explicitly designed for social interactions. Lord and Lomicka [33] focus on the opportunities distance learning offers for the creation of cross-institutional communities of practice. Their findings suggest that chats and face-to-face interactions promote a greater sense of community than do electronic discussion boards/internet forums. Here, too, the social dimension was made possible thanks to distance learning platforms. Similarly, Collopy and Arnold [34] indicate that face-to-face instruction clearly supports "team development, commitment and accountability to team members, and the processing of content with the instructor" (pp. 99–100), but the online space also allowed for "time to think, process, and have online conversations outside of scheduled class time" (p. 100). Overall, digital environments contributed to expanding the social dimension of learning. In all these studies, the online solutions are from an earlier decade, but they were already employed intentionally for specific purposes.

More recently, Bicen et al. [35] have argued that blended learning is more efficient than distance learning for multimedia projects in teacher education; their study thus documents the importance of in-person instruction for an effective learning design. This observation is related to the conclusions that Atmacasoy and Aksu [36] draw in their literature review: if blended learning contributes to a positive attitude towards the courses in general, face-to-face learning is generally favored due to the enhanced social interactions between peers and teachers. Chan [7] reinforced this result by tracking future teachers' preferences for traditional face-to-face lectures, interpreting it as being rooted in the pedagogical heritage of Confucian culture.

Other scholars study blended or online learning in teacher education programs, but focus on other matters, such as the advantages digital technologies have for teachers. Duhaney [37] looked into how blended learning can be used in teacher training courses, especially as a transition towards teacher implementation of ICT. Montgomery et al. [38] focused on the learning analytics made possible by online activities of students through a flipped blended learning approach in a teacher education program.

We are, however, unaware of any study that has experimentally investigated the experience of flow by students in teacher-training programs, that has endeavored to explain their preference for face-to-face learning environments, or that has detailed their perception of distance learning environments. This represents a clear research gap, especially as student engagement is considered to be one of the most important variables in academic achievement [39,40], and as a social dimension seems necessary for lifelong learning readiness [41]. Therefore, the objective of the present study is to contribute to instructional design for curricula in teacher education by collecting and assessing the self-reported learning experience of students in two synchronous environments (face-to-face and online). The goal is to support teacher education institutions in designing blended learning modules, or in choosing between the face-to-face and online learning environments, in order to offer an optimal learning experience to their students.

## 2. Materials and Methods

Given the research question, a specific learning context was chosen for an experiment with a detailed intervention involving face-to-face and online learning environments.

### 2.1. Context and Participants

French-speaking future secondary education teachers (24 women, 13 men, $M_{age}$ = 34.53 years, $SD_{age}$ = 7.51) attending modules during the second year of their three-year teacher-training program in Switzerland participated in the study. They voluntarily responded to questionnaires at the end of two separate one-week courses lasting 100 min.

All students were familiar with synchronous and asynchronous online learning environments, as the experiment occurred at the end of the fourth semester and all courses in the curriculum encompassed blended or distance learning. Moreover, the participants had previously attended four similar courses in a synchronous online learning environment using Adobe Connect with the same structure (see below: theory, mutual evaluation, and synthesis).

### 2.2. Intervention

The two courses were situated at the end of a semester-long learning sequence addressing the theory of pedagogical differentiation, the development of methods and materials, and the analysis of the implementation. The objective of the sequence is for students to work together to evaluate the differentiation practices implemented in class. The courses, held one week apart, were identical in objective and structure: (1) theoretical synthesis of the entire semester content—about differentiation in education—and group discussions as a question-and-answer session that lasted 20 min, (2) 50 min for group evaluation of differentiation practices, (3) synthesis of group work in a 10-minute session, and (4) completion of the research questionnaire for the study (20 min). The control setting was a

traditional face-to-face teaching unit in which the instructor interacted directly with each student. The experimental setting was in a synchronous online learning environment using Adobe Connect (see Figure 1), a web-based solution requiring only a browser and that is advertised as a versatile environment to design and bring to life virtual experiences for a diverse audience. Installed on up-to-date personal computers with a webcam and a microphone, Adobe Connect provides users with a chat section on the screen, a collective notetaking board, and an on-screen document-sharing function in addition to the video-conference system.

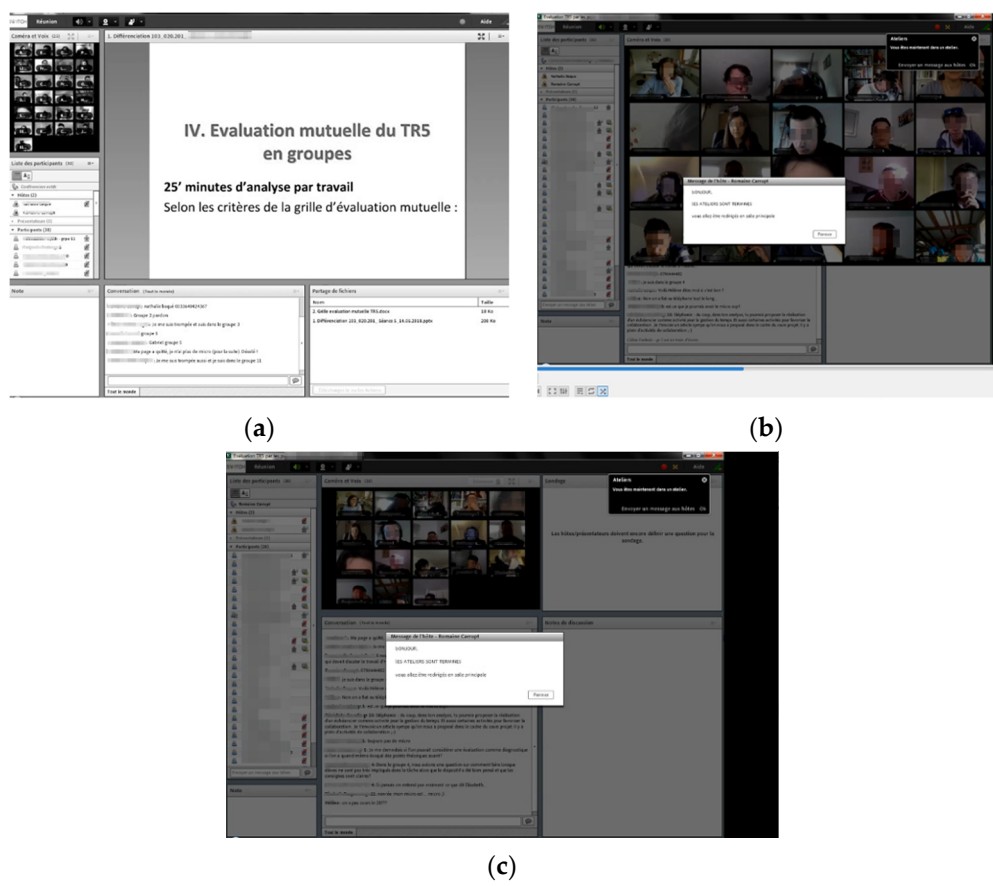

**Figure 1.** Screenshots of the online learning environment. Top left: (**a**) theory for the whole class; top right: (**b**) group work; (**c**) bottom: synthesis of group work in the whole class.

*2.3. Study Design*

As discussed in Li et al. [42], the number of participants included in this type of study ranges from 8 to 56 per group. In contrast to most of the studies referenced by Li et al., however, a repeated measures design was chosen for our study, which allows for a smaller sample size, as within-subjects design offers higher statistical validity [43]. In addition, the research project employed a counterbalanced design. During the first week, Group A ($n$ = 18) took the 100-minute course in the online learning environment and Group B ($n$ =19) took the 100-minute course in the traditional face-to-face setting in a standard classroom. One week later, the two groups switched formats, meaning that Group A experienced the traditional face-to-face setting and Group B the online learning environment (see Table 1). Both learning conditions were taught by the same teacher. By counterbalancing the type of environment, we controlled for the impact of time, which was not an interesting factor in our design.

**Table 1.** Organization of courses following the same structure.

| Group | T1 | T2 (One Week Later) |
|---|---|---|
| A | Online learning environment | Face-to-face environment |
| B | Face-to-face environment | Online learning environment |

Note. In each case, the course structure was as follows: (1) theory, (2) group evaluation, (3) synthesis of group work, (4) completion of the questionnaire.

### 2.4. Data Collection

The use of *EduFlow* [44], a self-reported measurement scale of flow experience, was selected for its compatibility with the research setting, its simplicity, its multidimensionality, and its validity in French, the language of the participants. It is a 12-item questionnaire using a 7-point Likert scale (1, "strongly disagree" to 7, "strongly agree") that assesses the four dimensions of optimal learning experience: cognitive absorption ("I feel that what I do is under my control"), immersion and time transformation ("I don't notice the time passing"), loss of self-consciousness ("I don't fear the judgment of others"), and autotelic experience ("I'm experiencing a moment of intense interest"). A new version of *EduFlow* was published [45] with some dimension names modified to better follow Csikszentmihalyi's recommendations. However, these modifications did not change the structural validity of the scale.

For each dimension, the mean of the three items constituting the dimension was retained, as proposed by the creators of the scale. Therefore, no specific coding procedures were necessary. These dimensions demonstrated suitable internal consistency in the present study, as seen in Table 2.

**Table 2.** Descriptive statistics for flow variables and according to measurement times.

| Variables | Face-to-Face [M(SD)] | Skewness | Kurtosis | Online Learning [M(SD)] | Skewness | Kurtosis | Cronbach Alpha |
|---|---|---|---|---|---|---|---|
| Cognitive control | 4.95 (1.27) | −0.888 | 0.603 | 4.47 (1.28) | −0.360 | 0.229 | 0.71 |
| Time transformation | 4.43 (1.27) | −0.354 | −0.264 | 3.76 (1.44) | 0.302 | −0.422 | 0.79 |
| Loss of self-consciousness | 4.30 (1.63) | −0.108 | −0.516 | 4.00 (1.67) | 0.693 | −0.263 | 0.83 |
| Autotelic experience | 3.14 (1.43) | 0.394 | −0.512 | 2.75 (1.43) | 1.34 | 1.51 | 0.82 |

## 3. Results

This section presents the data in a table and a figure, as well as the data analysis procedure.

### 3.1. Data Collected

The raw data files that gather time 1 and time 2 data for each group (A and B) are available on the Zenodo repository (http://doi.org/10.5281/zenodo.4550422, accessed on 16 May 2022). The descriptive statistics of the data collected are presented in Table 2, reporting mean and standard deviations as well as skewness and kurtosis indicators for each flow dimension in the face-to-face and online learning environments. The findings highlight that, except for autotelic experience in the online learning environment, the distribution of all variables appears to be relatively normal.

### 3.2. Data Analysis

Four repeated measures ANOVA, including the learning environment as a within-subject factor on each dimension of the flow, were performed using Jamovi software (Jamovi project, 2021). Sphericity was granted as we had two conditions in the within-subject factor.

The statistical results reveal that the learning environment has a significant effect on cognitive control ($F[1;36] = 9.68$, $p = 0.004$, $\eta^2_p = 0.212$), time transformation ($F[1;36] = 13.9$, $p < 0.001$, $\eta^2_p = 0.279$), and autotelic experience ($F[1;36] = 5.18$, $p = 0.029$, $\eta^2_p = 0.126$). Globally, the results suggest that face-to-face environments promote these dimensions of

optimal learning experience, as compared to online learning environments (see Figure 2). No further significant effect was found.

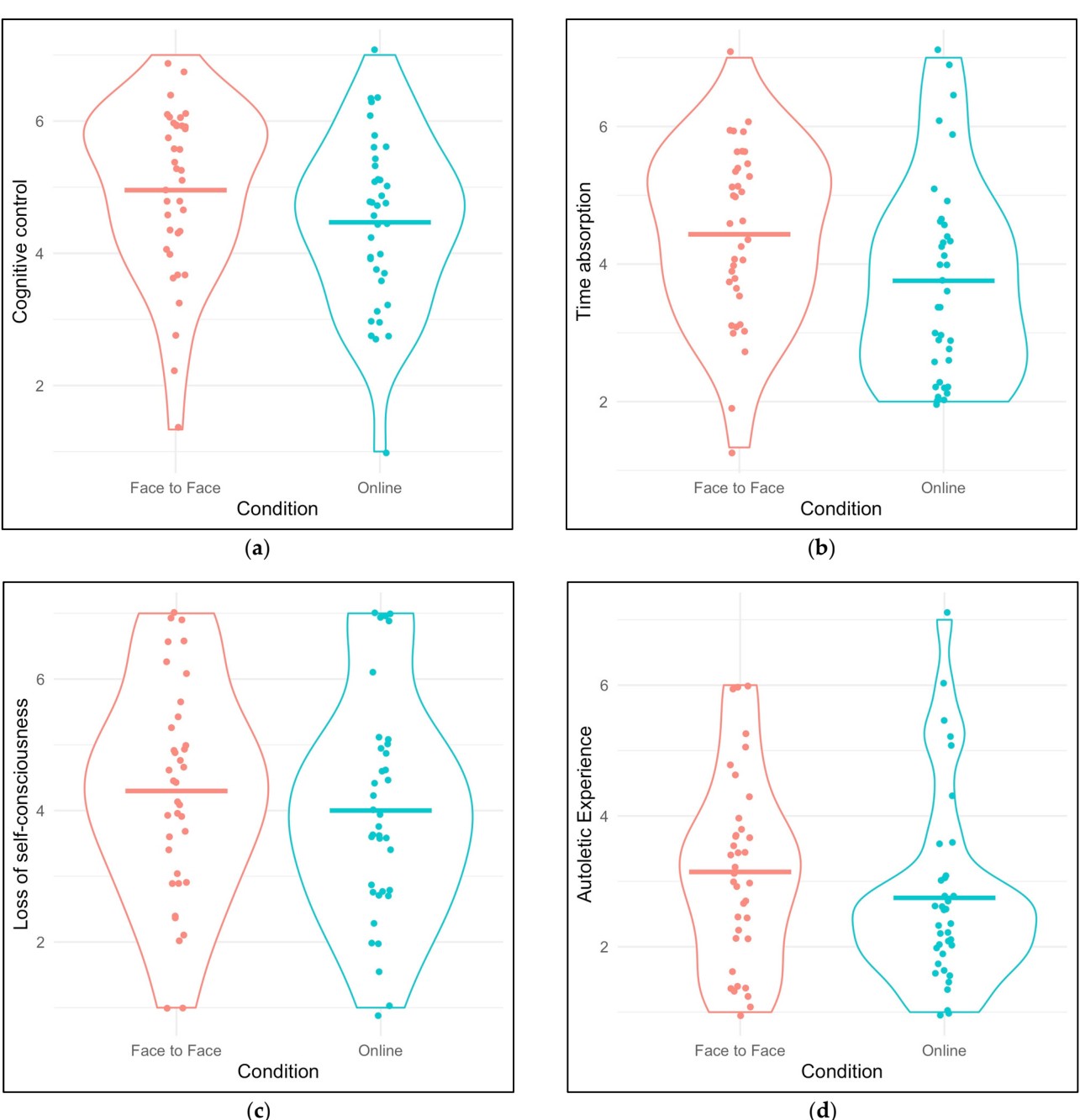

**Figure 2.** Main effect of the setting on (**a**) cognitive control, (**b**) time transformation, (**c**) loss of self-consciousness, and (**d**) autotelic experience, on a 7-point Likert scale.

## 4. Discussion

The overall aim of this counterbalanced study conducted using the *EduFlow* questionnaire was to assess the subjective learning experience (i.e., flow) in face-to-face vs. online environments, particularly as findings on this aspect are largely lacking. To date, very few experiments exploring the impact of blended learning in terms of affective and motivational issues in these contexts have been conducted, and the present study endeavors to contribute to reducing the knowledge gap in this area.

The findings suggest that participants experience greater flow when learning in a face-to-face setting in comparison to an online environment. More precisely, even after becoming accustomed to distant or blended learning over the past two years due to the pandemic, the future teachers reported higher levels of cognitive absorption, time transformation, and autotelic experience when taking the same course face-to-face rather than online. The self-consciousness dimension of the *EduFlow* questionnaire was the only dimension with similar results in both learning settings.

### 4.1. Impacts of Face-to-Face and Online Environments on Self-Reported Flow

First, the data collected indicate no effect concerning loss of self-consciousness (Table 2). This is possibly related to the fact that the students were familiar with such blended learning systems, that they knew each other quite well—this study took place at the end of a semester-long learning sequence—and they were therefore comfortable working together, both in the face-to-face setting and the online environment. Moreover, the tasks were similar in both sequences, and no psychosocial factor in relation to self-consciousness was changed: in each environment, students worked in small groups of three to five, where they were required to interact and collaborate on a common task related to their practical teaching activity. This result indicates that the online environment or the computer interface did not inhibit communication or social exchanges. In other words, the reported interactions were experienced similarly in both settings. This initial result suggests that if students have experience with distance learning and know each other, online classes (in groups or the whole class) that are related to their future professional activity can accommodate (or at least do not preclude) the psycho-social dimension of learning.

However, the results also showed that three other dimensions of flow—perception of time, concentration and immersion in the tasks, and well-being while learning—were significantly higher in the face-to-face setting than in the online learning environment. Thus, even for students who are experienced in online settings and during a particularly meaningful activity, attention seems to decrease in distance learning or online lessons. This confirms and possibly further expands the findings of Herbert et al. [5] and their conclusion about higher engagement when learning occurs in class, face-to-face. The lack of spontaneity of interactions may influence these variables even among students accustomed to working online. In a distance learning environment, we need to switch on the microphone, find a window of opportunity to speak, and sometimes deal with glitches in the internet connection that diminish the experience. These factors may reduce both student attention and motivation, even if they have experience in these settings. Moreover, the benefits of face-to-face learning environments recall the concept of natural pedagogy [46], which states that meeting in person has an impact on student attention and learning because teachers are fully connected, fine-tuning their movements, and perpetually regulating their activities according to learning cues given by the learners. In adult education, where discussions are important for the learning experience, this barrier to fine-tuning in interaction could pose a barrier to experiencing flow. In addition, the present findings reinforce Alvarez and Steiner's [18] conclusions about the challenges that add complexity to the online environments for learning and that demand specific technical, cultural, and pedagogical skills. It is key that instructional design for the digital environment encompasses specific parameters and, consequently, it is important that it does not simply transfer what was designed for face-to-face education to the online setting.

The lower immersion reported in the online learning environment may also be related to the medium in the pedagogical relationship. Indeed, a machine comes between the humans who are interacting, and this digital interface comes with multiple distractions [47,48]—whether regarding academic or work-related activities (e.g., e-mailing, web searches) or leisure (e.g., social media, video games)—that lead students to divide their attention among several tasks and that, per se, may reduce flow. After all, students are just a click away from playing a game or performing other unrelated tasks such as shopping or online chatting. This increased simultaneous engagement in two or more activities may, of

course, decrease concentration and immersion in the task and make students feel less involved or absorbed in the lesson. After all, an effort must be made to participate in learning, and this demands resources: when attending class online, it is critical to avoid distractions and to stay focused on the lesson. To help students in such online situations, the teacher must bolster motivation and immersion in the task through mechanisms that capture and maintain attention. However, the findings of the present study suggest that providing meaningful activities and group tasks to students who already know each other and who are familiar with such online pedagogical settings is not adequate to promote immersion in distance learning. Thus, even when such measures are adopted, face-to-face learning may still bring about a deeper cognitive engagement, partly because it has fewer distractors.

Finally, it is possible to question the advantage of the face-to-face environment for autotelic experience and for the changes in student perception of how time passes. Indeed, when designing the learning environment, it may be problematic per se to seek to improve cognitive well-being in students, to increase their desire to engage in an activity again, or to change how students perceive the passing of time. This would be the case if, for instance, the learning experience is intentionally conceived to be disruptive to cognitive equilibrium [49]. In other words, there are situations in which complexity or difficulty should be anything but an autotelic experience in which a person loses track of time.

### 4.2. Limitations and Pitfalls

The following limitations to this study should be noted. First, the results must be considered with caution, as the sample size is relatively small ($n = 37$) and given the fact that the majority of the sample (65%) is composed of one gender (women). Even if the counterbalanced design helps counteract this aspect, this factor could nonetheless impact the generalization of the results. Moreover, post hoc achieved power was computed for each significant effect. The results reveal that the only variable reaching a power above 0.80 was time transformation (power = 0.84; power for cognitive absorption = 0.73 and power for autotelic experience = 0.38). This suggests that our results need to be replicated with a bigger sample size. Second, it may be necessary to refer to other specific influences on motivation and attention to rigorously compare face-to-face and online learning environments in teacher education programs. For instance, the students in our study may have no particular interest in computer-mediated teaching, as they are engaged in a training program for professional practice in schools where distance learning is not the norm. In addition, the measurements were obtained only on the basis of self-reported questionnaires. Further research is therefore needed to confirm the findings, particularly experiments using more objective measurement methods, such as physiological measures of stress and engagement, academic results, direct observation of student attention, time spent on a task, or the duration of participation. Third, the data collected were used to assess the entire intervention ((1) theoretical synthesis, (2) group evaluation of practice, (3) synthesis of group work). It is possible that the learning experience would have been reported differently if each section had been assessed individually. As flow represents a highly focused and productive period of active engagement, it may not be a completely suitable variable for studying and evaluating an entire course. For instance, we could hypothesize that the theoretical section alone would have brought fewer contrasting results. Thus, future research projects in teacher education should focus on specific teaching situations (e.g., listening vs. speaking/working on tasks, one-on-one settings, small groups, large groups) and compare the experience of flow in the face-to-face and the distance online environments in these various situations.

Based on all these considerations, the results need to be replicated in different teaching and learning contexts, and additional data must be collected to more clearly accommodate learning environment design. For instance, the flow experienced by students may have been judged differently in both settings had there been a different teaching focus. Moreover, in the case of a content-based or a teacher-driven lecture, it is conceivable that the environment—whether face-to-face or online—would have little influence, while dif-

ferent results might be expected in a student-based classroom setting. Second, gathering observational data during the teaching and learning phase in the place of self-reported data after the learning phase would help identify teaching contexts and tasks that promote flow. Third, combining flow data and actual learning data according to each setting would be of great benefit for decision making in learning environment design. Finally, blending face-to-face and online learning environments and assessing flow could generate useful data to support instructional design in teacher education.

## 5. Conclusions

Distance learning environments are common at universities. However, teacher education programs, in particular, need to be designed on evidence-based interventions so that teaching practice models can be reproduced in the classroom. To date, research findings on how face-to-face vs. online pedagogical environments impact flow among future teachers are largely lacking. In this study, we examined the self-reported flow of students in teacher-training programs in two different learning settings: face-to-face and synchronous online environments. The counterbalanced research design allowed for documenting the learning experience by using the *EduFlow* questionnaire that each student completed. Unlike other studies on blended or distance learning that focus on performance or satisfaction, this experiment documented flow and the four dimensions of an optimal learning experience.

The results show a significant impact of the face-to-face setting on three of four dimensions of flow. Globally, the findings suggest that the in-person environment promotes cognitive absorption, time transformation, and autotelic experience. Other significant differences were not ascertained. It is possible to understand the better flow experience in the face-to-face environment compared to the distance learning environment as being related to the importance of direct social interactions for future teachers, as proposed by Donnelly [32]; it may also be related to digital barriers [18], which some teachers might have difficulty in overcoming. By contrast, however, and given that participants of the present study were familiar with working in distance learning environments, the proposition of a cultural habit may be invalid [7].

The findings also point to the benefits of introducing supportive measures when designing digitized learning environment for students in teacher-training programs. Students seem to experience greater flow and retain their attention better in traditional, face-to-face classrooms, but online learning environments may promote time efficiency or scalability. If gamification or specific instructional design is added to the digital learning experience, it is possible that the reported flow would attain the levels in the face-to-face setting. For now, however, without a specific design for the distance online environment that promotes variability of learning experiences, it may be beneficial for the learners' experience of flow to remain in a face-to-face design. Overall, this study provides a starting point for the documentation of flow experience among future teachers, an important topic that could help universities of teacher education improve their curricula. Nevertheless, further studies are still needed to encourage or reject specific forms of online learning environments and to precisely identify when they are useful.

**Author Contributions:** Conceptualization, L.A., R.C., C.A. and P.G.; methodology, P.G., R.C. and C.A.; software, C.A.; validation, L.A., R.C., C.A. and P.G.; formal analysis, C.A.; investigation, R.C.; resources, R.C.; data curation, C.A.; writing—original draft preparation, L.A.; writing—review and editing, L.A., R.C., C.A. and P.G.; visualization, C.A.; supervision, L.A.; project administration, L.A. All authors have read and agreed to the published version of the manuscript.

**Funding:** This research received no external funding.

**Institutional Review Board Statement:** Ethical review and approval were waived for this study due to informed consent obtained from all participants as well as to the fact that the learning experience was standard and approval by the research institutions.

**Informed Consent Statement:** Informed consent was obtained from all subjects involved in the study.

**Data Availability Statement:** Raw data files are accessible on Zenodo repository (http://doi.org/10.5281/zenodo.4550422, accessed on 16 May 2022).

**Conflicts of Interest:** The authors declare no conflict of interest.

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
