# Peer review of "Self-Reported Flow in Online Learning Environments for Teacher Education: A Quasi-Experimental Study Using a Counterbalanced Design"

_education, doi:10.3390/educsci12050351_

Round 1

Reviewer 1 Report

The manuscript presents the results of a study whose problematic relevance is of great contemporary relevance, in particular following the COVID 19 pandemic.

However, some points need to be revised.

1) Although the authors have fully indicated this in the limits of this study, it would be wise to check whether, given the small number of people involved,  all the statistical treatments are relevant, in particular possibly via G*Power by Faul and colleagues (2007, 2009).

2) Although the literature review presented in the first part of the paper is interesting, it would be worthwhile to focus on flow and optimal experience in the field of education (e.g. Heutte, 2020 ; Shernoff & Csikszentmihalyi, 2009) and technology (e.g. Riva, Baños, , Botella,  Wiederhold, & Gaggioli, 2012).

This literature review could also be updated, in particular by taking inspiration from the references cited in the scoping review of Flow Research by Peifer and colleagues (2022).

3) the authors use the Eduflow model, which is quite judicious, but unfortunately, they have used a version that is not up to date.

Indeed, the version used in the chapter (Heutte, Fenouillet, Kaplan, Martin-Krumm & Bachelet, 2016) is in fact the first version of the Eduflow model (Heutte, Fenouillet, Boniwell, Martin-Krumm, & Csikszentmihalyi, 2014) which was updated on the opportunity of a presentation at ECPP 2016, EduFlow-2 (Heutte, Fenouillet, Martin-Krumm, Boniwell & Csikszentmihalyi, 2016) and whose final publication took place at the end of last year (Heutte et al. 2021).

Indeed, there was a conceptual problem in the first version of Eduflow (Heutte, Fenouillet, Boniwell, Martin-Krumm, & Csikszentmihalyi, 2014) : the absence of the immersion component in dimension FlowD1 named ‘Cognitive absorption’ had been pointed. Further research demonstrated that immersion was in fact associated with FlowD2. Consequently, this dimension has been renamed ‘Immersion and Time transformation’ (Heutte, Fenouillet, Martin-Krumm, Boniwell & Csikszentmihalyi, 2016). Following Csikszentmihalyi’s recommendation, the name of FlowD1 was updated, without modifying the items, to more precisely describe this sub-dimension: « In any case, a sense of control is definitely one of the most important components of the flow experience, whether or not an ‘objective’ assessment justifies such feelings » (Csikszentmihalyi, 1975, p. 46). FlowD1 is now named ‘Cognitive control’, nevertheless the previous research remains valid as the structure of FlowD1 has not been modified.

As only one item (in FlowD2) out of the 12 has been modified in EduFlow-2, the vast majority of the data remains valid, however, FlowD1 and FlowD2 need to be renamed correctly.

References

Faul, F., Erdfelder, E., Buchner, A., & Lang, A.-G. (2009). Statistical power analyses using G*Power 3.1: Tests for correlation and regression analyses. Behavior Research Methods, 41, 1149-1160.

Heutte, J. (2020). Psychologie positive et formation des adultes : le flow ou le plaisir de comprendre tout au long de la vie. Savoirs, 3, 17-61.

Peifer, C., Wolters, G., Heutte, J., Tan, J., Freire, T., Tavares, D., ... & Triberti, S. (2022). A Scoping Review of Flow Research. Frontiers in Psychology, 12 1-27. https://doi.org/10.3389/fpsyg.2022.815665

Riva, G., Baños, R.-M., Botella, C., Wiederhold, B.-K., Gaggioli, A. (2012) Positive technology: using interactive technologies to promote positive functioning. Cyberpsychology, behavior and social networking, 15(2), 69-77

Author Response

  • Thanks for the constructive feedback.
  • The proposed literature was examined and introduced in the article when it added value.
  • The statistical concerns were reexamined and an addition has been added to the text. ??????????
  • A reference to the new version of EduFlow and its difference with the previous version has been added to the text

Reviewer 2 Report

The paper is well written and well argumented. The used methods are sound. The limits of the performed research are also well discussed. I have no objection to the publication of this paper.

Author Response

  • Thanks for the review.
  • A spell check has been conducted and some modifications occurred.

Reviewer 3 Report

The author(s) examine future secondary education teachers’ experiences of flow, which is an optimal learning experience, in online versus face-to-face courses. They collect students’ self-reported questionnaire answers and use a four-dimensional model that consist of cognitive absorption, time transformation, loss of self-consciousness and well-being to identify their experiences of flow throughout online and face-to-face course settings.

In the method section, it is not clear that whether both online and face-to-face courses were taught by the same instructor or not. If the authors provide information about it, it will be helpful to interpret the findings of the study. If the online and face-to-face classes were taught by different teachers, their different level of teaching experience in online and face-to-face settings can enable students in face-to-face settings experience more optimal learning experiences than their peers in online version of the course.

The authors emphasize that the field of study has an influence on students’ experience of learning in online activities (line 130-131, p. 3). However, they do not provide adequate information about the content of the course students take in online and face-to-face settings. Is it a pedagogical course to teach a certain subject (e.g., math, science, history)? The author(s) just mentioned about the general structure of the course. They state that the “objective of the course sequence is for student work together and evaluate the differentiation practices implemented in class.” (line 197 – 198, p. 4). It is not clear what the authors mean by saying “differentiation practices.” If they provide more specific information about the course content, it will be helpful.  

The table 2 splits between the page 6 and 7. If they put the whole table into one page, it will be helpful to interpret the results presented on the table.

Also, if the authors put the questionnaire on the appendix of the study, it will be helpful for the readers to determine whether the items in the questionnaire reflect students’ experiences of flow and other optimal learning experiences.

If the author(s) revise the paper based on the detailed feedback provided above, it will enhance the quality of the paper and its contribution to the research literature.

Author Response

  • Thanks for the constructive feedback.
  • A short precision has been added in relation to the content of the teaching in the chapter "Intervention." However, we do not provide a definition of what is "differentiation in education", because it is complex, multifaceted, and the object of several courses within the teacher-training program.
  • Regarding the availability of the questionnaire used, we decided not to include it, as EduFlow2 is published and freely available here https://doi.org/10.3389/fpsyg.2021.828027.

Round 2

Reviewer 1 Report

In order to avoid conceptual confusion (better follow Csikszentmihalyi’s recommendations), linked to one of the differences in the naming of the dimensions of the EduFlow scale (v1, 2014 ==> v2, 2021), it would be better to rename FlowD1 as "cognitive control" (instead of "cognitive absorption").

On the other hand, there is an error concerning the authors of reference 44
https://www.persee.fr/doc/stice_1764-7223_2014_num_21_1_1110

Author Response

Thanks for the feedback. Your proposition has been integrated into our document.